

# Prevalence and risk factors of recurrent aphthous stomatitis among college students at Mangalore, India

Matthew Antony Manoj[1], Animesh Jain[2], Saanchia Andria Madtha[1] and
Tina Mary Cherian[1]

[1] Kasturba Medical College, Mangalore, Manipal Academy of Higher Education, Manipal, India
[2] Department of Community Medicine, Kasturba Medical College, Mangalore, Manipal Academy of Higher
Education, Manipal, Karnataka, India

## ABSTRACT

**Background.** Recurrent aphthous stomatitis (RAS) is one of the most common oral mucosal diseases affecting an approximate 25% of the world's population. Some common etiological factors are genetics, nutritional deficiencies, stress and immune dysfunction. There is currently no specific medication to treat the condition but RAS tends to heal by itself within a week or two. We aimed to explore about the prevalence and related risk factors of recurrent aphthous ulcers among college students aged 18–30 years who had been affected within the preceding six months prior to the study duration.

**Methods.** A questionnaire survey was conducted among 681 students from four colleges in Mangalore, Karnataka, India after obtaining the approval for the same from the respective colleges. Consenting participants returned a survey containing various questions. The collected data was then analyzed using descriptive statistics. The study was approved by the Institutional Ethics Committee.

**Results.** Of the 681 participants, 322 (47.2%) were affected with RAS in the past six months which included 131 (40.6%) males and 191 (59.3%) females. Single mouth ulcers were the most common presentation seen among the study participants (74.2%). Factors showing statistically significant association were: family history of RAS ($P <$ 0.001), known diabetics ($P < 0.001$), history of smoking ($P < 0.001$), oral trauma ($P < 0.001$), history of wearing braces/dentures ($P < 0.001$) as well as those using toothpastes containing sodium lauryl sulphate ($P < 0.001$), stress and lack of sleep ($P < 0.001$). The most common form of medication used were topical agents (43.1%) ($P < 0.001$).

**Conclusions.** There was a statistically significant association between the occurrence of RAS and family history of RAS, diabetes, smoking, history of braces/dentures, oral trauma, sodium lauryl sulphate toothpastes, lack of sleep, stress, menstruation, consumption of particular foods and beverages. Further research is needed in this field to truly understand the prevalence and risk factors of RAS and to help in discovering a treatment modality for this condition.

Corresponding author
Animesh Jain, animesh_j@yahoo.com

## INTRODUCTION

Recurrent Aphthous Ulcer (RAU) or recurrent aphthous stomatitis or more commonly known as mouth ulcers affects an approximate 25% of the world's population (*Shulman, Beach & Rivera, 2004*). Studies have reported the prevalence of RAS in the United States and United Kingdom to be about 20% of the population (*Santosh et al., 2014*). With a variety of factors contributing to the cause of RAS neither a specific risk factor, nor a treatment has been discovered.

Various factors such as gender, body mass index (BMI), age, family history, wearing dental braces, gastrointestinal diseases, times of brushing, bedtime at night, are some of the factors that contribute to RAS (*Qian, Shenglou & Liu, 2018*). It has also been suggested that the occurrence of recurrent aphthous stomatitis can result from oral trauma or even from the consumption of acidic foods and carbonated drinks (*Crispian, 2006*). Mouth ulcers lasting more than 3 weeks could be suggestive of mouth tumors leading to orofacial pain and soreness (*Crispian & Rosemary, 2001*). RAS has been reported to affect the patient's lifestyle quality, however, studies have reported that RAS did not cause anxiety or depression among patients (*Abdalwahab, 2015*). Stress was found out to be one of the main etiological factors of RAS (*Usha, Sunita & Vaishali, 2017*). Tobacco users were found to have a less likely chance of being affected with RAS than non-tobacco users (*Shamaz & Chandrashekar, 2014*). It was hypothesized that patients suffering from recurrent aphthous stomatitis were having vitamin B1, vitamin C, iron and/or calcium deficiencies (*Ogura et al., 2001*). There has not yet been any correlation established between RAS and its genetic factor (*Miller et al., 1977*). Several local, systemic, immunologic, allergic, and microbial factors as well as immunosuppressive drugs have been proposed as causative agents (*Sunday & Martin, 2014*). A study conducted in Tamil Nadu, India shows that the main cause of RAS could be systemic factors, food allergy or even due to immunopathogenesis. The study also showed that 17% of the population were affected with Minor RAS (*Namrata & Abilasha, 2017*).

RAS usually heals itself within one week or two. There are no medications yet discovered that will help cure RAS completely but local anesthetics such as topical gels, creams, ointments, mouth wash can be used. It was also found out that taking vitamin supplements will help prevent and cure RAS (*Sook & Stephen, 1996*).

This is the only study done in Karnataka, India which was designed to investigate the prevalence of RAS among college students aged 18–30 years in the Indian population and to assess the independent risk factors associated with RAS. Through this study, we aim to contribute towards research into the topic of aphthous stomatitis.

## MATERIALS AND METHODS

A cross-sectional descriptive study was carried out in Mangalore, Karnataka among individuals aged 18-30 years who have or have not been affected with RAS. Four institutions comprising of a medical college, dental college, and two other degree colleges were selected as the study setting. The study was conducted from March 1st, 2019 to May 30th, 2019. The sample size of 681 was calculated by taking 20% relative precision, 95% confidence, 15% prevalence (*Santosh et al., 2014*) and 20% non-response. The participants were chosen
on the basis of a non-random sampling. Individuals who have or have not been affected with mouth ulcers and those who had complete questionnaires were included in the study. Incomplete questionnaires and those who did not accept the informed consent was excluded from the study. The study protocol was submitted to the Institutional Ethics Committee of Kasturba Medical College (Manipal Academy of Higher Education), Mangalore (IEC KMC MLR 02-1963). After obtaining the approval of the ethics committee, approval was taken from the heads of the respective colleges. After obtaining the approval of the colleges, data collection was done using the structured survey questionnaire through online Google Forms. An informed consent form was obtained from every individual. In the first section of the questionnaire, demographic data of the participants was recorded. In subsequent section, participants were asked about whether they had a history of being affected by aphthous stomatitis ever and/or within the past six months, family history of aphthous stomatitis and about the location and pattern of the aphthous ulcers. Participants were also asked certain questions about their lifestyle like dietary choices, smoking habits, sleeping habits, and brushing habits. In addition to this, participants were also asked about the history of dental braces usage, history of diabetes, nutritional deficiencies, systemic diseases and about the treatment they sought for the aphthous stomatitis.

**Data analysis**

The data were entered into MS Excel and tabulated. Data were analyzed using descriptive statistics. A statistical package SPSS version 25.0 was used to do the analysis. The chi-square test was used to analyze the association between the occurrence of recurrent aphthous stomatitis in the participants and family history of aphthous stomatitis, history of diabetes, and smoking habits. In addition to this, the chi-square test was also used to analyze an association between the occurrence of aphthous stomatitis among the participants and history of dentures, oral trauma, type of toothpaste used, sleep habits, stress, menstruation, and dietary habits. $P < 0.05$ was considered to be statistically significant.

## RESULTS

Of the total 681 participants that participated in the study, 268 (39.4%) of the participants were male and 413 (60.6%) of the participants were female, giving a male to female ratio of 2:3. The prevalence of RAS among individuals affected in the past six months prior to the study period was 47.2%.

Of the total 681 participants in this study, 322 (47.2%) suffered from RAS in the past six months of which 131 (40.6%) were males and 191 (59.3%) were females which was significant (Table 1). Each participant was given the option to choose multiple responses to help establish the pattern and anatomical location of the ulcers among participants that were affected by it. Of the total 322 individuals that have been affected by RAS in the past six months, 239 (74.2%) reported single mouth ulcers, 25 (7.7%) reported the presence of a cluster of ulcers and 58 (18%) reported a combination of both which was significant (Table 2). The preeminent anatomical locations of the ulcers were found to be on the inner surface of the lips with a prevalence of 78.2% followed by the inner surface of the cheeks

**Table 1  Participants affected by recurrent aphthous stomatitis (RAS) in the past six months.**

| Gender | Affected by RAS in the past six months | | | | | | | | $\chi^2$ | P value |
|---|---|---|---|---|---|---|---|---|---|---|
| | Yes | | No | | Not applicable | | Total | | | |
| | N | % | N | % | N | % | N | % | | |
| Male | 131 | 40.6 | 94 | 34.2 | 43 | 51.2 | 268 | 39.4 | | |
| Female | 191 | 59.3 | 181 | 65.8 | 41 | 48.8 | 413 | 60.6 | 8.252 | 0.016 |
| Total | 322 | 100.0 | 275 | 100.0 | 84 | 100.0 | 681 | 100.0 | | |

**Table 2  Pattern and anatomical location of mouth ulcers present in participants in the past six months.**

| Pattern of mouth ulcer in the past six months | | | |
|---|---|---|---|
| Pattern | N | % | |
| Single mouth ulcers | 239 | 74.2 | $P \leq 0.001$ |
| Clusters | 25 | 7.7 | |
| Both | 58 | 18 | |
| Anatomical location of mouth ulcer in the past six months | | | |
| Location | N | % | |
| Dorsal surface of tongue | 89 | 27.6 | |
| Inner surface of lips | 252 | 78.2 | $P \leq 0.001$ |
| Inner surface of cheeks | 189 | 58.6 | |
| Ventral surface of tongue | 43 | 13.3 | |
| Soft palate | 13 | 4.0 | |

with a prevalence of 58.6% and on the dorsal surface of the tongue giving a prevalence of 27.6% which was also significant (Table 2).

A total of 254 (37.2%) participants reported a family history of recurrent aphthous stomatitis of which 190 (59%) suffered from RAS in the past six months while 51 (18.5%) participants who had a family history of RAS did not suffer from RAS in the past six months which was significant (Table 3). Out of 28 diabetic patients who participated in the study, 24 participants were affected with RAS in the past six months (Table 3). Of a total 176 participants who were active smokers, 118 (36.6%) were affected with RAS in the past six months. From the total study population of 322 affected with RAS in the past six months, 111 (34.4%) participants who stopped smoking still suffered from RAS during the past six months which was significant (Table 3). There was a total of 142 individuals who stopped smoking of which 111 were affected with RAS which was also very significant (Table 3).

Out of a total of 251 (36.8%) participants reported a history of wearing braces/dentures, 156 (48.4%) participants who had reported a history of wearing dentures/braces suffered from RAS (Table 4). It was also discovered that 224 (69.5%) suffered from RAS in the past six months due to trauma of the oral mucosa which was significant (Table 4). A total of 272 (84.4%) participants who actively used toothpastes containing sodium lauryl sulphate were affected with RAS in the past six months which was also very significant (Table 4).

It was also observed that 199 (61.8%) participants who suffered from RAS, slept for five to six hours and 94 (29.1%) participants who were affected with RAS slept for seven

**Table 3  Role of family history of RAS, diabetes, smoking and cessation of smoking among individuals who were affected with RAS in the past six months.**

| Family history of RAS | Affected by RAS in the past six months | | | | | | | | $X^2$ | P value |
|---|---|---|---|---|---|---|---|---|---|---|
| | Yes | | No | | Not applicable | | Total | | | |
| | N | % | N | % | N | % | N | % | | |
| Yes | 190 | 59.0 | 51 | 18.5 | 13 | 15.4 | 254 | 37.2 | | |
| No | 132 | 39.7 | 224 | 81.4 | 71 | 84.5 | 427 | 62.7 | 123.33 | $\leq 0.001$ |
| Total | 322 | 100.0 | 275 | 100.0 | 84 | 100.0 | 681 | 100.0 | | |
| **Diabetes** | Affected by RAS in the past six months | | | | | | | | $X^2$ | P value |
| | Yes | | No | | Not applicable | | Total | | | |
| | N | % | N | % | N | % | N | % | | |
| Yes | 24 | 7.4 | 3 | 1.0 | 1 | 1.1 | 28 | 4.1 | 31.442 | $\leq 0.001$ |
| No | 298 | 92.5 | 272 | 98.9 | 83 | 98.8 | 653 | 95.8 | | |
| Total | 322 | 100.0 | 275 | 100.0 | 84 | 100.0 | 681 | 100.0 | | |
| **Smokers** | Affected by RAS in the past six months | | | | | | | | $X^2$ | P value |
| | Yes | | No | | Not applicable | | Total | | | |
| | N | % | N | % | N | % | N | % | | |
| Yes | 118 | 36.6 | 46 | 16.7 | 12 | 14.2 | 176 | 25.8 | 40.253 | $\leq 0.001$ |
| No | 204 | 63.3 | 229 | 83.2 | 72 | 85.7 | 505 | 74.1 | | |
| Total | 322 | 100.0 | 275 | 100.0 | 84 | 100.0 | 681 | 100.0 | | |
| **Smoking Cessation** | Affected by RAS in the past six months | | | | | | | | $X^2$ | P value |
| | Yes | | No | | Not applicable | | Total | | | |
| | N | % | N | % | N | % | N | % | | |
| Yes | 111 | 34.4 | 29 | 10.5 | 2 | 2.3 | 142 | 20.8 | | |
| No | 13 | 4.0 | 35 | 12.7 | 4 | 4.7 | 52 | 7.6 | 96.14 | $\leq 0.001$ |
| Not Applicable | 198 | 6.1 | 211 | 76.7 | 78 | 92.8 | 487 | 71.5 | | |
| Total | 322 | 100.0 | 275 | 100.0 | 84 | 100.0 | 681 | 100.0 | | |

to eight hours daily which was highly significant (Table 4). Among the participants who were affected with RAS in the past six months, 126 (39.1%) participants reported that they suffered from RAS during stressful periods of their life which was also significant (Table 4).

Of the total 191 female participants who were affected with RAS in the past six months, 58 (18%) female participants reported that they were affected with aphthous stomatitis specifically during menstruation which was significant (Table 4).

Pineapples (29.1%) were the most common fruit to be foremost cause for the formation of ulcers followed by lemons (14.9%) and oranges (10.2%) but this was not statistically significant (Table 5). However, it was also reported that 38.5% of the participants attributed spicy food to be the fundamental cause for the formation of ulcers followed by chocolates (9.0%) and almonds (5.2%) (Table 5). Due to the consumption of beverages, 54 (16.7%) participants accredited coffee to be a major cause in the formation of ulcers followed by pineapple juice and orange juice which was reported by 37 (11.4%) and 35 (10.8%) participants respectively (Table 5).

It was observed that a total of 424 participants consumed carbonated beverages frequently. Of the total 322 participants who were affected with RAS in the past six months, 162 (50.3%) participants who consumed carbonated beverages one-two times
**Table 4** Role of braces/dentures history, oral trauma, use of toothpaste containing sodium lauryl sulphate, sleep, stress and menstruation among participants affected with RAS in the past 6 months.

| History of braces/dentures | Affected by RAS in the past six months | | | | | | | | $X^2$ | P value |
|---|---|---|---|---|---|---|---|---|---|---|
| | Yes | | No | | Not applicable | | Total | | | |
| | N | % | N | % | N | % | N | % | | |
| Yes | 156 | 48.4 | 82 | 29.8 | 13 | 15.4 | 251 | 36.8 | | |
| No | 166 | 51.5 | 193 | 70.1 | 71 | 84.5 | 430 | 63.1 | 40.940 | ≤ 0.001 |
| Total | 322 | 100.0 | 275 | 100.0 | 84 | 100.0 | 681 | 100.0 | | |

| Oral trauma | Affected by RAS in the past six months | | | | | | | | $X^2$ | P value |
|---|---|---|---|---|---|---|---|---|---|---|
| | Yes | | No | | Not applicable | | Total | | | |
| | N | % | N | % | N | % | N | % | | |
| Yes | 224 | 69.5 | 131 | 47.6 | 9 | 10.7 | 364 | 53.4 | | |
| No | 98 | 30.4 | 144 | 52.3 | 75 | 89.2 | 317 | 46.5 | 221.54 | ≤ 0.001 |
| Total | 322 | 100.0 | 275 | 100.0 | 84 | 100.0 | 681 | 100.0 | | |

| Toothpaste containing sodium lauryl sulphate | Affected by RAS in the past six months | | | | | | | | $X^2$ | P value |
|---|---|---|---|---|---|---|---|---|---|---|
| | Yes | | No | | Not applicable | | Total | | | |
| | N | % | N | % | N | % | N | % | | |
| Yes | 272 | 84.4 | 218 | 79.2 | 70 | 83.3 | 560 | 82.2 | | |
| No | 50 | 15.5 | 57 | 20.7 | 14 | 16.6 | 121 | 17.7 | 18.093 | 0.006 |
| Total | 322 | 100.0 | 275 | 100.0 | 84 | 100.0 | 681 | 100.0 | | |

| Sleep | Affected by RAS in the past six months | | | | | | | | $X^2$ | P value |
|---|---|---|---|---|---|---|---|---|---|---|
| | Yes | | No | | Not applicable | | Total | | | |
| | N | % | N | % | N | % | N | % | | |
| Less than 5 Hours | 24 | 7.4 | 19 | 6.9 | 1 | 1.1 | 44 | 6.4 | | |
| 5 to 6 Hours | 199 | 61.8 | 116 | 42.1 | 14 | 16.6 | 329 | 48.3 | | |
| 7 to 8 Hours | 94 | 29.1 | 126 | 45.8 | 53 | 63.0 | 273 | 40.0 | 98.422 | ≤ 0.001 |
| More than 8 Hours | 5 | 1.5 | 14 | 5.0 | 16 | 19.0 | 35 | 5.1 | | |
| Total | 322 | 100.0 | 275 | 100.0 | 84 | 100.0 | 681 | 100.0 | | |

| Stress | Affected by RAS in the past six months | | | | | | | | $X^2$ | P value |
|---|---|---|---|---|---|---|---|---|---|---|
| | Yes | | No | | Not applicable | | Total | | | |
| | N | % | N | % | N | % | N | % | | |
| Yes | 126 | 39.1 | 51 | 18.5 | 6 | 7.1 | 183 | 26.8 | | |
| No | 196 | 60.8 | 224 | 81.4 | 78 | 92.8 | 498 | 73.1 | 64.207 | ≤ 0.001 |
| Total | 322 | 100.0 | 275 | 100.0 | 84 | 100.0 | 681 | 100.0 | | |

| Menstruation | Affected by RAS in the past six months | | | | | | | | $X^2$ | P value |
|---|---|---|---|---|---|---|---|---|---|---|
| | Yes | | No | | Not applicable | | Total | | | |
| | N | % | N | % | N | % | N | % | | |
| Yes | 58 | 18.0 | 16 | 4.9 | 0 | 0 | 74 | 10.8 | | |
| No | 133 | 41.3 | 165 | 51.2 | 41 | 48.8 | 439 | 64.4 | 58.621 | ≤ 0.001 |
| Not Applicable | 131 | 40.6 | 94 | 29.1 | 43 | 51.1 | 168 | 24.6 | | |
| Total | 322 | 100.0 | 275 | 100.0 | 84 | 100.0 | 681 | 100.0 | | |

every week were affected with RAS followed by 34 (10.5%) participants who drank carbonated beverages three-four times every week and 31 (9.6%) participants who consumed carbonated beverages daily which was highly significant (Table 6). There

**Table 5  Dietary factors among participants affected with RAS in the past six months.** This table lists the probable dietary factors among participants affected with RAS in the past six months as reported by themselves.

| Fruits | Affected by RAS in the past six months | | | | | | | |
| | Yes | | No | | Not applicable | | $X^2$ | $P$ value |
| | N | % | N | % | N | % | | |
|---|---|---|---|---|---|---|---|---|
| Pineapple | 96 | 29.1 | 53 | 19.2 | 1 | 1.1 | | |
| Apple | 21 | 6.5 | 7 | 2.5 | 1 | 1.1 | | |
| Fig | 15 | 4.6 | 7 | 2.5 | 0 | 0 | | |
| Lemon | 48 | 14.9 | 15 | 5.4 | 1 | 1.1 | 80.189 | 0.083 |
| Orange | 33 | 10.2 | 13 | 4.7 | 2 | 2.3 | | |
| Strawberry | 10 | 3.1 | 6 | 2.1 | 0 | 0 | | |
| Banana | 10 | 3.1 | 4 | 1.4 | 0 | 0 | | |
| NA/None | 185 | 57.4 | 200 | 72.7 | 79 | 94.0 | | |
| **Food** | **Affected by RAS in the past six months** | | | | | | | |
| | Yes | | No | | Not applicable | | $X^2$ | $P$ value |
| | N | % | N | % | N | % | | |
| Chocolate | 29 | 9.0 | 8 | 2.9 | 0 | 0 | | |
| Cheese | 16 | 4.9 | 5 | 1.8 | 0 | 0 | | |
| Peanut | 16 | 4.9 | 3 | 1.0 | 0 | 0 | | |
| Almond | 17 | 5.2 | 6 | 2.1 | 1 | 1.1 | 107.010 | ≤ 0.001 |
| Egg | 15 | 4.6 | 10 | 3.6 | 0 | 0 | | |
| Spicy Food | 124 | 38.5 | 52 | 18.9 | 2 | 2.3 | | |
| NA/None | 157 | 48.7 | 200 | 72.7 | 81 | 96.4 | | |
| **Beverages** | **Affected by RAS in the past six months** | | | | | | | |
| | Yes | | No | | Not applicable | | $X^2$ | $P$ value |
| | N | % | N | % | N | % | | |
| Coffee | 54 | 16.7 | 25 | 9.0 | 0 | 0 | | |
| Tea | 22 | 6.8 | 7 | 2.5 | 0 | 0 | | |
| Milk | 16 | 4.9 | 3 | 1.0 | 0 | 0 | | |
| Carbonated Drinks | 31 | 9.6 | 8 | 2.9 | 1 | 1.1 | 90.971 | 0.004 |
| Orange Juice | 35 | 10.8 | 5 | 1.8 | 0 | 0 | | |
| Pineapple Juice | 37 | 11.4 | 37 | 13.4 | 2 | 2.3 | | |
| NA/None | 190 | 59.0 | 219 | 79.6 | 1 | 1.1 | | |

was no significant relation between individuals who were frequent consumers of coffee (Table 6), tea (Table 6) and milk (Table 6) with the recurrence of aphthous stomatitis.

It was reported that when affected with aphthous ulcers, 139 (43.1%) individuals used topical agents such as triamcinolone acetonide, fluocinolone acetonide, clobetasol propionate, topical diclofenac and topical lidocaine (Table 7). Vitamin supplements were reported to be used by 65 (20.1%) participants while 51 (15.8%) participants resorted to home remedies (Table 7). Oral medications such as hydrocortisone buccal tablets and prednisolone tablets were used by 30 (9.3%) participants (Table 7). Mouth washing agents such as chlorhexidine gluconate was used by 42 (13%) participants. It was also reported that 89 (27.6%) participants did not take any sort of medication (Table 7).

**Table 6 Relationship between RAS and frequency of consumption of beverages among participants affected in the past six months.** This table depicts the relationship between RAS and frequency of consumption of beverages among participants affected in the past six months

| Consumption of carbonated beverages | Affected by RAS in the past six months | | | | | | | | | |
| --- | --- | --- | --- | --- | --- | --- | --- | --- | --- | --- |
| | Yes | | No | | Not applicable | | Total | | $X^2$ | P value |
| | N | % | N | % | N | % | N | % | | |
| Nil | 95 | 29.5 | 124 | 45.0 | 38 | 45.2 | 257 | 37.7 | | |
| Daily | 31 | 9.6 | 14 | 5.0 | 8 | 9.5 | 53 | 7.7 | | |
| 1–2 times a week | 162 | 50.3 | 112 | 40.7 | 29 | 34.5 | 303 | 44.4 | 20.846 | 0.002 |
| 3–4 times a week | 34 | 10.5 | 25 | 9.0 | 9 | 10.7 | 68 | 9.9 | | |
| Total | 322 | 100.0 | 275 | 100.0 | 84 | 100.0 | 681 | 100.0 | | |

| Consumption of coffee | Affected by RAS in the past six months | | | | | | | | | |
| --- | --- | --- | --- | --- | --- | --- | --- | --- | --- | --- |
| | Yes | | No | | Not applicable | | Total | | $X^2$ | P value |
| | N | % | N | % | N | % | N | % | | |
| Nil | 168 | 52.1 | 145 | 52.7 | 50 | 59.5 | 363 | 53.3 | | |
| Daily | 3 | 0.9 | 0 | 0 | 1 | 1.1 | 4 | 0.5 | | |
| 1–2 times a week | 139 | 43.1 | 122 | 44.3 | 31 | 36.9 | 292 | 42.8 | 11.752 | 0.163 |
| 3–4 times a week | 12 | 3.7 | 8 | 2.9 | 2 | 2.3 | 22 | 3.2 | | |
| Total | 322 | 100.0 | 275 | 100.0 | 84 | 100.0 | 681 | 100.0 | | |

| Consumption of tea | Affected by RAS in the past six months | | | | | | | | | |
| --- | --- | --- | --- | --- | --- | --- | --- | --- | --- | --- |
| | Yes | | No | | Not applicable | | Total | | $X^2$ | P value |
| | N | % | N | % | N | % | N | % | | |
| Nil | 198 | 61.4 | 176 | 54.6 | 48 | 57.1 | 422 | 61.9 | | |
| Daily | 0 | 0 | 2 | 0.6 | 1 | 1.1 | 241 | 35.3 | | |
| 1–2 times a week | 117 | 36.3 | 92 | 28.5 | 32 | 38.0 | 15 | 2.2 | 4.961 | 0.549 |
| 3–4 times a week | 7 | 2.1 | 5 | 1.5 | 3 | 3.5 | 3 | 0.4 | | |
| Total | 322 | 100.0 | 275 | 100.0 | 84 | 100.0 | 681 | 100.0 | | |

| Consumption of milk | Affected by RAS in the past six months | | | | | | | | | |
| --- | --- | --- | --- | --- | --- | --- | --- | --- | --- | --- |
| | Yes | | No | | Not applicable | | Total | | $X^2$ | P value |
| | N | % | N | % | N | % | N | % | | |
| Nil | 138 | 42.8 | 110 | 40 | 36 | 42.8 | 284 | 41.7 | | |
| Daily | 3 | 0.9 | 6 | 2.1 | 2 | 2.3 | 11 | 1.6 | | |
| 1–2 times a week | 163 | 50.6 | 143 | 52 | 38 | 45.2 | 344 | 50.5 | 5.269 | 0.728 |
| 3–4 times a week | 18 | 5.5 | 16 | 5.8 | 8 | 9.5 | 42 | 6.1 | | |
| Total | 322 | 100.0 | 275 | 100.0 | 84 | 100.0 | 681 | 100.0 | | |

**Table 7 Various medications used for RAS by participants in the past six months.** Medications used by participants in the past six months for relief from RAS are depicted in this table.

| Medication | Affected by RAS in the past six months | | | | | |
| --- | --- | --- | --- | --- | --- | --- |
| | Yes | | No | | $X^2$ | P value |
| | N | % | N | % | | |
| Topical Agents | 139 | 43.1 | 49 | 17.8 | | |
| Oral Medication | 30 | 9.3 | 12 | 4.3 | | |
| Vitamin Supplements | 65 | 20.1 | 29 | 10.5 | 168.486 | $\leq 0.001$ |
| Mouth Wash | 42 | 13.0 | 22 | 8.0 | | |
| Home Remedies | 51 | 15.8 | 35 | 12.7 | | |
| NA/None | 89 | 27.6 | 160 | 58.1 | | |

## DISCUSSION

Recurrent aphthous stomatitis proves to be challenge to physicians all around the world due to its difficulty in diagnosis and treatment. RAS continues to be a topic of research in the fields of medicine, otorhinolaryngology, dermatology and oral surgery. Studies from all around the world emphasize on the fact that it is important for the professionals of the medical world to detect the clinical aspects of this condition because every patient is treated differently. RAS remains a common problem faced by college students with various factors such as stress, lack of sleep, consumption of beverages, and smoking contributing to its cause. To our knowledge, this is the first study of its kind among college students in Karnataka, India.

In the present study conducted among individuals between the ages of 18–30 years in Mangalore, Karnataka, females constituted a larger proportion of the total study population affected with RAS in the past six months. Similar results have been reported in studies conducted in the northern and southern parts of India. In a study conducted by *Mathew et al. (2008)*, it was reported that the prevalence of RAS was 2.1% in southern India. In another study conducted by *Patil et al. (2014)*, it was reported that the prevalence of RAS in northern India was found to be 21.7%. A study held in Maharashtra reported that only 72 patients among 71,851 patients were clinically diagnosed with aphthous stomatitis reporting a prevalence of 0.1% (*Rajmane et al., 2017*). These statistics indicates that the prevalence of RAS varies among the general population of India. Certain epidemiological studies have reported that the prevalence of aphthous stomatitis among the adult population in the United States and Canada was 46.4% to 69.4% (*Embil, Stephens & Manuel, 1975*). The reported prevalence of aphthous stomatitis in Europe was 36% to 37% (*Embil, Stephens & Manuel, 1975*). A study held in Sweden reported that the prevalence of aphthous stomatitis in Sweden was 0.5% to 2% (*Robledo-Sierra et al., 2013*).

In our study, the prevalence of RAS was more common among the female participants. In the study conducted by *Patil et al. (2014)* it has been reported that there was a higher prevalence among females (56.3%) than in males (43.7%). It has been suggested by some authors that hormonal factors could be responsible for the higher prevalence of RAS among females (*Ship et al., 2000*). On the contrary, only a few studies have reported a higher prevalence of RAS among males (*Okoh & Ikechukwu, 2019*).

The inner surface of the lips was the preeminent location for the occurrence of aphthous stomatitis (78.2%) followed by inner surface of the cheeks (58.6%) and lastly on the dorsal surface of the tongue (27.6%). In a study held in Rio by *Queiroz et al. (2018)* it was reported that the most common anatomical location for the occurrence of RAS was the tongue followed by the buccal mucosa. In this study, it was also discovered that the most common form of presentation was single mouth ulcers (74.2%) followed by a cluster of ulcers (7.7%) and lastly a combination of both (18%).

An earlier study by *Ship (1965)* found that RAS had a definite tendency to occur along family lines and that the probability of a sibling developing RAS was influenced by the parents RAS status. The results in our study have proved the same, 59.0% of the individuals who were affected with RAS six months prior to the study period reported a family history

of recurrent aphthous stomatitis which was statistically significant ($P \leq 0.001$). More recent investigations have detected associations between RAS and specific HLA subtypes, which indicates that RAS in certain individuals may have a genetic basis (*Ship et al., 1965*).

The presence of oral mucosal lesions such as RAS has been frequently diagnosed in patients with diabetes mellitus. The actual prevalence is rarely discussed in clinical studies. In a study by *Silva et al. (2015)*, the investigators identified a high prevalence of RAS among diabetic patients (78.4%). In this study a total of 28 diabetic patients had participated of which 24 were affected with RAS in the past six months which was statistically significant ($P \leq 0.001$). These results go on to highlight the importance for both physicians as well as dentists to closely monitor diabetic patients for oral mucosal lesions.

The results of this study showed a significant occurrence of aphthous ulcers in smokers and in former smokers. A total of 176 individuals who participated in the study were active smokers of which 118 (67%) were affected with RAS in the past six months which was significant ($P \leq 0.001$) and a total of 142 individuals were former smokers of which 111 (78.1%) participants were affected with RAS in the past six months which was also significant ($P \leq 0.001$). Our results are in line with the theory that aphthous ulcers are common in smokers as tobacco causes injury or chronic irritation to the oral mucosa (*Tuzun et al., 2000*). The results in this study about former smokers being affected with RAS are in line with previous studies that have reported that aphthous ulcers are common in former smokers with a possibility that certain former smokers have a chance to develop severe laceration (*McRobbie, Hajek & Gillison, 2004*). Studies have debated that the use of tobacco especially smoking has a "protective effect" on aphthous ulceration. It has been suggested that smokers have an increased keratinization of the oral mucosa and this keratinization provides a protects the oral mucosa against trauma and bacterial penetration (*Axéll & Henricsson, 1985*). It is possible that one of the absorbed constituents that promotes keratinization may be hyperkeratosis. Although hyperkeratosis may be a premalignant condition, it is possible that it prevents aphthous ulcers through a local protective effect on the oral mucosa (*Sawanir, 2010*). In addition to this, it is not clear whether it is the nicotine presence in the tobacco that induces the protective effect or whether it is the presence of one of the other constituents (*Rivera-Hidalgo, Shulman & Beach, 2004*). The theory that nicotine is the protective factor, is supported by a recent report that aphthous ulcers were prevented among nonsmokers with recurrent aphthous ulcers while they used nicotine gum (*Shapiro, Olson & Chellemi, 1970*).

Trauma such as sharp food, dental procedures, braces/dentures, traumatic tooth brushing, etc. have been suggested as one of the common causes of aphthous ulcers (*Ship, 1996*). Similarly, in this study it was observed that there was a significant association of RAS occurrence with oral trauma and in those individuals with a history of braces/dentures ($P \leq 0.001$). Certain studies have reported that individuals who use toothpaste's containing sodium lauryl sulphate are more susceptible to aphthous ulcers (*Herlofson & Barkvoll, 1994*). Our results are in line with these studies showing that 272 participants of a total 560 participants who used toothpastes containing sodium lauryl sulphate were affected with RAS in the past six months which was significant ($P = 0.006$).

We believe that stress influences the duration of RAS rather than trigger it (*Huling et al., 2012*). A total of 39.1% of the participants in our study reported that they were affected with RAS when they were faced with stress in their life ($P \leq 0.001$). In this study, it was also observed that 223 (69.2%) participants who were affected with RAS in the past six months slept for less than six hours daily which was significant ($P \leq 0.001$). A similar study by *Ma et al. (2015)* reported that bedtime after 11pm was an independent risk factor for aphthous ulcer recurrence among college students.

Certain studies have reported that there is a correlation between menstruation and aphthous ulcers (*Thangadurai et al., 2015*) but there are also studies that state that there is not enough evidence to support this claim (*Preeti et al., 2011*). It was observed in our study that there was a statistically significant association between the occurrence of RAS and menstruation ($P \leq 0.001$).

According to the results in the current study, 29.1% of the participant's reported pineapples as the foremost cause for the formation of ulcers followed by lemons (14.9%) and oranges (10.2%) ($P = 0.083$) (Table 5). It has been suggested by certain authors that glycosides in the pineapple stimulates the oral mucosa and the presence of protease leads to allergic reactions in some people (*Zhao, 2015*). Many studies have reported spicy food as one of the most common causes for the occurrence of aphthous ulcers. Spicy foods result in a temporary shortage of free moisture in the mouth due to high calories (*Bao et al., 2015*). Reduction of saliva in the mouth fails in protection of oral mucosa. Our findings are in line with this theory as spicy food was reported as the most common type of food associated with the occurrence of RAS which was statistically significant ($P < 0.001$) (Table 5). Following the consumption of coffee, 16.7% of the participants reported that they were affected by ulcers followed by pineapple juice (11.4%) and orange juice (10.8%) ($P = 0.004$) (Table 5). Coffee contains a substance called theobromine which is sensitive to the oral region. The habits of sweet and acidic intake can lead to changes in pH of the mouth. It is reported that stomatitis is more likely to occur when pH in the mouth is abnormal (*Tian et al., 2002*). It was also discovered in this study that 50.3% of the participants who drank carbonated drinks one-two times a week were affected with RAS in the past six months followed by 10.5% of the participants who consumed carbonated beverages three-four times a week which was very significant ($P = 0.002$) (Table 6). A study held by *Du et al. (2018)*, showed similar results that frequent consumption of carbonated beverages was an independent risk factor for RAS. It was observed that the prevalence of RAS was higher in individuals who were frequent consumers of carbonated beverages (*Du et al., 2018*). Carbonated drinks soften the enamel surface leading to extremely rough, porous, and alveolate demineralization, which then causes the wear of soft tissues in the mouth (*Gambon, Brand & Nieuw Amerongen, 2010*). There was no association found between RAS and frequent consumers of tea, milk and coffee.

In this study, it was reported that the most common treatment used was topical agents followed by vitamin supplements and home remedies (Table 7). It is commonly reported that the treatment for aphthous ulcers is symptomatic mainly with the use of topical agents. Systemic therapy should only be considered in patients with chronic aphthous stomatitis (*Altenburg et al., 2014*).
## CONCLUSION

The prevalence of aphthous ulcers within six months preceding study was found to be 47.2% among college students at Mangalore. There was a statistically significant association between the occurrence of RAS and family history of RAS, diabetes, smoking, history of braces/dentures, oral trauma, sodium lauryl sulphate toothpastes, lack of sleep, stress, menstruation, consumption of particular foods and beverages. This research would add on to the current evidence and help in the prevention of ulceration. Further research is needed in this field to truly understand the prevalence and risk factors of RAS and to help in discovering a treatment modality for this condition.

### Funding
The authors received no funding for this work.

### Competing Interests
Animesh Jain is an Academic Editor of PeerJ. The other authors declare that they have no competing interests.

### Author Contributions

- Matthew Antony Manoj conceived and designed the experiments, performed the experiments, analyzed the data, prepared figures and/or tables, authored or reviewed drafts of the article, and approved the final draft.
- Animesh Jain conceived and designed the experiments, analyzed the data, authored or reviewed drafts of the article, and approved the final draft.
- Saanchia Andria Madtha conceived and designed the experiments, performed the experiments, prepared figures and/or tables, authored or reviewed drafts of the article, and approved the final draft.
- Tina Mary Cherian conceived and designed the experiments, performed the experiments, prepared figures and/or tables, authored or reviewed drafts of the article, and approved the final draft.

### Human Ethics
The following information was supplied relating to ethical approvals (i.e., approving body and any reference numbers):

The study protocol was approved by the Institutional Ethics Committee of Kasturba Medical College, Mangalore (IEC KMC MLR 02-1963).

### Data Availability
The raw data is available in the Supplemental Files.

## Supplemental Information

Supplemental information for this article can be found online at http://dx.doi.org/10.7717/peerj.14998#supplemental-information.

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

## FURTHER READING

**Scully C, Porter S. 1989.** Recurrent aphthous stomatitis: current concepts of etiology, pathogenesis and management. *Journal of Oral Pathology and Medicine* **18**:21–27 DOI 10.1111/j.1600-0714.1989.tb00727.x.