# Peer review of "Prevalence and risk factors of recurrent aphthous stomatitis among college students at Mangalore, India"

_PeerJ, doi:10.7717/peerj.14998_

## Round 0.1 · original submission · Major Revisions

Dear author,

The manuscript in its current form has major flaws which need reworking with it. Especially in the areas of statistical analysis and interpretation. Also, the manuscript needs language correction.

Reviewer 1 ·

Basic reporting

The article is clear and unambiguous, English need to be reviewed (foe example; mouth wash, yet) and authors need to give attention to capitalization throughout the manuscript (for example; Page 3 line 35, Mouth; Page 4, line 70; Questionnaire..).

The number of references is enough however more papers has to be cited specially in the prevalence section (nationwide and internationally) to show the prevalence of this disorder in relation to other target groups.
Professional article structure, figures, tables: there are to many tables (7) (it could be merged!)
The presentation of the tables is not professional.

Experimental design

The manuscript is within the aims and scope of the journal (Medical Sciences, and Health Sciences).
Research question well defined, relevant & meaningful. It is stated how research fills an identified knowledge gap (the prevalence of RAS is unknown in the group).
Rigorous investigation performed to a high technical & ethical standard: (ethical approval shown (page4 line 73).
Methods described with sufficient detail but we need more information on how tested the allergy from food and how to measure the stress/anxiety levels? or they just reported based on direct questioning from the participants?).

Validity of the findings

Impact and novelty not assessed. Meaningful replication encouraged where rationale & benefit to literature is clearly stated (findings address the prevalence of RAS in this specific group and this well help to understand the behavior of this condition and may help later on to analyze the data on a bigger context).

All underlying data have been provided.
Conclusions are well stated, linked to original research question & limited to supporting results.

Additional comments

Authors has to employ the proper name of this condition (Recurrent aphthous stomatitis RAS) throughout the manuscript and not using other names such as RAU.
There is a mixed up between RAS and aphthous-like ulcer?
The relation between RAS and trauma? (there is a subtype, trauma-induced RAS!)
Its to high to have all these individuals with braces/denture among this group!
BMI (P: 3, line 39 (body mass index)
P: 3, line; 44 (its proven that RAS is affecting the QoL!)
P: 3, Line 48; iron and/or calcium
P: 3, Line 49 and 54 yet!
Table 2; RAS not affecting hard palate! (usually its affecting non-keratinized mucosa.
Number of diabetes in this group is high!
Too many factors were employed in the analysis of this paper!, its too long.
Authors claims that many things (such as food, P: 10, line; 133; cause formation of ulcers!)
Table 7; what are the topical agents, oral medications and mouthwashes used?
Table 7: the total and adding up to 100?, numbers not matching
Discussion:
Line: 176, 177 please add the studies show RAS is more common among males.
Line 185; Ship (a single author, there is no coworkers (no et al, REF#18).
Line 200, 201 Please review the percentage (36.6! its 118/176?)
Line; 202 111/142 (78.1%)
There are many references show the protective affects of the tobacco against the RAS!!

·

Basic reporting

The authors have tried to address an emerging yet silent public health concern through this novel research work. The language used for preparing this scientific manuscript is clear throughout except in few places which is suggested below :
1. Line 19 and 20 - Results section of abstract : "single mouth ulcers were the most common" do the authors mean to say it as the most common presentation among the RAU seen among study subjects. Kindly make the necessary corrections.
2. Line 180 - spelling mistake "Togue" should be corrected as "tongue"
3. Line 179 - remove the first "the" from "......and lastly the on the dorsal..."
4. Line 193 - change "diabetic" to diabetes

Review of literature is comprehensive , article structure is as per the journal requirements.

Experimental design

Research question is clear and the study design is apt for addressing the knowledge gap however it can be made clear to the readers from the scientific community that the study design was "cross sectional descriptive study" which was done and not just cross sectional.

Validity of the findings

Results are well summarised with descriptive and inferential statistics however the value "Not applicable" under table 1,3,4,5 is not clear. Even the responses under not applicable is not constant in each of the tables (N=84 in majority and in few NA is 79, 81 and so on). To simplify the table interpretation "N" can be summarised or defined as sub script below the table for variables/data.
table 2 : "anatomical location"- define the responses if there were multiple responses taken for each participant as "N" is not clear.
Elaborating the findings of the results in discussion again can be avoided and emphasis can be given on agreements or reason for different findings in other studies.

·

Basic reporting

This paper explored the prevalence and related risk factors of recurrent aphthous ulcers among college students aged 18-30 years who had been affected within the preceding six months prior to the study duration. The results of the article have implications for the field of research on recurrent aphthous ulcers. There are also major issues with data analysis and interpretation throughout the paper.

Experimental design

1.For the introduction section: Authors should clarify the definition and distinction between recurrent aphthous ulcers and oral ulcers. When referring to this concept in the paper, it should be unified.

2.For the statistic analysis:
The article not only used the methods of descriptive statistics, but also used other statistical methods, such as the chi-square test, which should be explained in detail in the methods section. The authors should also clarify how the chi-square and p-values for Table 4-7 were derived. In addition, multivariate analysis with logistic regression is recommended to analyze the risk factors.

3.The authors mentioned in the introduction section that "Various factors such as gender, BMI, age, family history, wearing dental braces, gastrointestinal diseases, times of brushing, bedtime at night, are some of the factors that contribute to RAU". However, BMI and age were not fully analyzed in the follow-up analysis. If these data are collected, they should be analyzed. If not, the reasons should be explained.

4.In the Methods section, it should be clearly stated how many people completed the questionnaire, and the number and proportion of incomplete questionnaires. It is recommended to draw a flowchart.

Validity of the findings

5.Recurrent aphthous ulcers are usually divided into three types, including mild, severe, and herpetiform. The interval and healing time of recurrent aphthous ulcers are important indicators of its severity, so data on interval and healing time should be collected and analyzed. For this article, if no relevant data have been collected, a discussion of the deficiencies should be included in the discussion section of the article.

---

## Round 0.2 · Minor Revisions

Dear author,
I feel the manuscript still needs some information and clarity in the methodology section. As the reviewer has pointed out, the use of statistical tests should be elaborated, preferably as a separate section. Also, there is also a need to describe the questionnaire that was used with appropriate references and details of validation. As it was mentioned that the questionnaire was distributed using Google forms, explain the various sections of the questionnaire and the questions that were included in each section in brief.

·

Basic reporting

Structure of the article is comprehensive and clear with a thorough review of literature

Experimental design

Research question is clear and the methodology is clear after rereview

Validity of the findings

The limitations suggested by the authors are well understood when it comes to statistical analysis, however as it is corrected for multiple responses and entry by respondents the readers should not have difficulty in interpreting it now.

Additional comments

No comment

·

Basic reporting

no comment

Experimental design

For the statistic analysis:
The article not only used the methods of descriptive statistics, but also used other statistical methods, such as the chi-square test, which should be described in detail in the methods section.

Validity of the findings

no comment

---

## Round 0.3 · accepted · Accept

Dear author,

Based on the review reports and revisions made, your manuscript is accepted for publication.

·

Basic reporting

no comment

Experimental design

no comment

Validity of the findings

no comment

Additional comments

no comment